# Optimising older adults' home spaces to enhance their physical activity level: an exploratory qualitative study protocol

Naureen Akber Ali Meghani  ,[1] Joanne Hudson,[1] Gareth Stratton,[1] Jane Mullins[2]

¹Applied Sports Technology, Exercise and Medicine (A-STEM) Research Centre, Swansea University College of Engineering - Bay Campus, Swansea, UK
²College of Human and Health Sciences, Haldane Building, Singleton Park Campus, Swansea University, Swansea, UK

**Correspondence to**
Naureen Akber Ali Meghani; 2132644@swansea.ac.uk

## ABSTRACT

**Introduction** Housing is a major contributing factor to health, and better housing condition has been linked to improved general and psychological health. There has also been strong evidence that the physical environment within the home setting substantially impacts sedentary behaviour and physical activity in children. However, there is a lack of research that has investigated the physical environment within the home setting in the context of older adults' physical activity levels and sedentary behaviour. Given that with increasing age, older adults spend large proportions of their time in their homes it is important to optimise older adults' home settings to support healthy ageing. Therefore, this study aims to explore older adults' perceptions around optimising their physical environment within their home space to support physical activity and subsequently facilitate healthy ageing.

**Methods and analysis** In this formative research, a qualitative exploratory research design using in-depth interviews (IDIs) and a purposive sampling approach will be employed. IDIs will be conducted to collect data from study participants. Older adults from diverse community organisations in Swansea, Bridgend and Neath Port Talbot will formally request approval to recruit via its contacts for this formative research. The study data will be analysed thematically using NVivo V.12 Plus software.

**Ethics and dissemination** Ethical approval for this study has been obtained from the College of Engineering Research Ethics Committee (NM_31-03-22), Swansea University. The findings of the study will be disseminated to the scientific community and to the study participants. The results will enable us to explore the perceptions and attitudes of older adults towards physical activity within their home environment.

## BACKGROUND

Old age is defined as the most advanced stage of the biological cycle that is characterised by the manifestation of physical degradation and a general weakening of the organism. It is often linked with the state of inactivity and, subsequently, of need, as well as an excessive fragility from a health perspective.[1] However, the extension of life expectancy due to scientific advancement means that it is essential to view old age from a wider perspective, taking into account that it is not only a phase in which mental and physical impairments might

occur but also as an invaluable moment that can take on unique meaning.[2] In a similar vein, older adults face transitions and change common to this phase of life, such as children leaving the home, retirement, the passing of friends and loved ones, sexuality and bodily changes, and the occurrence of illnesses.[3] When it comes to ageing, the experience of space, especially the home space, becomes very important. Therefore, it is possible to plan and construct this space so that people will occupy it optimally and their home space will reflect and reinforce their identity.[4] A qualitative analysis of old age, time and the home space revealed that irrespective of loneliness, older adults expressed an emotional attachment with their living spaces, its role in connecting the memory of the past and life in general, reflecting a counter narrative to the stereotypical view of old age as characterised mainly by vulnerability.[5]

## STRENGTHS AND LIMITATIONS OF THIS STUDY

⇒ A well-established qualitative reflexive thematic analysis approach is used to understand in depth older adults' perception towards their physical activity within the home space.
⇒ A socialecological framework (Stokols, 1992) that is based on physical aspects of the home will be used in the analysis of interviews.
⇒ All semistructured interviews will be conducted either face-to-face or online via Zoom or WhatsApp call depending on the older adult's preference . The call feature may restrict the researcher's ability to observe non-verbal cues during interviews.

Older adults appear to have a greater sense of meaning in life than younger people. Moreover, a research also showed that human relationships were noted as an important factor for strength and meaning in life among older adults with good cognitive functioning. Furthermore, engaging in new activities that provided meaning in life after retirement was more likely among high cognitive functioning older adults. The study

also added that feeling worthless, along with poor health and physical functioning were significant contributors to a sense of meaninglessness of life among older people.[6] A Korean study conducted on 198 older adults revealed strong relationships between meaning in life and health, depression and problematic life events.[7] Hence, health is an important factor for meaning in life and consequently for healthy ageing.

Similarly, generativity (such as feeling useful and needed)[8] can be considered as one indicator of a healthy ageing process.[9] A study including older adults from diverse cultural backgrounds (German, Czech, Hong Kong Chinese and Cameroonian) showed a positive impact of generative concern on meaning in life, which was partially mediated by generative goals, after controlling for education level, competence, cognitive function and relationship.[10] The perceptions of older adults surrounding key aspects of generativity (such as feeling useful and needed) may also be related to later-life health trajectories. In older Finnish birth cohorts, not feeling needed by others was associated with a higher chance of death and institutionalisation (over a period of 5–10 years).[11] Similar results were found in rural-living French (60 years and above) and Japanese (65 years and above) older adults. Those with low perceptions of social usefulness experienced increased impairment in activities of daily living or died throughout follow-ups of 4–6 years.[12–14] In sum, higher perceptions of generativity are linked to older persons' longer lifespans and better physical performance over time.[8] Therefore, it is important to improve physical functioning and general health and well-being among older adults. Evidence suggests that physical activity (PA) is a modifiable behaviour linked to health, improved physical functional status and longevity, and promoting a physically active lifestyle has become a recognised public health objective.[15 16]

PA is a key factor of healthy ageing; however, many older adults are not sufficiently physically active and have sedentary lifestyles, which leads to detrimental health consequences, and even places financial and social constraints on themselves and society.[17] This is a global concern, with longevity increasing around the world, it is estimated that the proportion of older adults in the global population is expected to nearly double from roughly 12%–22% between 2015 and 2050, translating to an increase from 900 million to 2 billion people over the age of 60.[18] However, it is important to note that the aforementioned increase in life expectancy may not necessarily imply an increase in healthy life expectancy, as older people can face a variety of health issues.[19] These include cognitive function decline, reduced functional movement, impaired mental health and chronic conditions such as diabetes, high blood pressure and cardiovascular disease.[19] These health issues can have an impact on an older person's quality of life,[20] and increase healthcare costs and societal burden (eg, medical care expenses).[21] According to estimates, physical inactivity contributes to the worldwide burden of non-communicable diseases

(NCDs), with healthcare systems costs attributed to inactivity estimated at INT$53.8 billion globally, and further costs linked with INT$13.7 billion in productivity losses.[22 23] Reducing the mortality rates attributable to NCD calls for considerable action and equitable access to effective and good quality preventative and curative care, including behaviour modification such as increasing PA.[24] In that regard, engagement in activity (as part of leisure, sport, work and transportation) plays a significant role in successful ageing.[25–27] Thus, developing strategies that are tailored to meet the needs of different subgroups (eg, older adults) and that account for cultural contextual influences on behaviours related to, and attitudes towards, healthy ageing is paramount. Doing so will enable us to identify strategies to combat the effects of NCDs, improve and preserve functional capacity, and promote physical and mental health among older persons to support independence and improve quality of life.[27 28] Therefore, there is an urgent need to increase older adults' PA levels and decrease their sedentary behaviour.

The WHO has placed a particular emphasis on the promotion of PA to support healthy ageing.[29] The Global Action Plan on Physical Activity 2018–2030 identifies the strategic goal of increasing PA opportunities to engage inactive populations such as older adults.[30] Of concern, studies have shown that some older adults perceive PA to be only beneficial to the younger population[31]; however, there is compelling evidence that the requirement for PA does not end in later-life.[32] Doing so has numerous benefits, including a positive influence on the ability to perform daily living activities,[33 34] and findings from longitudinal studies demonstrate that engaging in regular PA can improve cognitive ability[32]lts exhibited much lower arterial stiffness as compared with their sedentary peers (standardised mean difference −1.017±0.340, 95% CI −1.684 to −0.350, p=0.003). Further, there is a drastic decline in the risk of cardiovascular disease mortality from 25% to 40% that accompanies low and high doses of moderate to vigorous PA, respectively. PA is related to a 12% decline in the risk of cancers, when comparing people with the highest level with those with the lowest level of activity. Review findings also suggest that inactive older people are at greater risk of all-cause mortality, fractures, repeated falls and functional restriction as compared with their active peers.[35] Thus, physical inactivity imposes detrimental consequences on older adults' physical well-being.

Physical inactivity also has negative effects on psychological well-being. It has been observed that PA has a considerable effect in reducing the mental health related burden among older adults.[32 36 37] According to the WHO (2017), it is estimated that around 20% of older adults (60 years and above) suffer from a mental health or neurological disorder that impacts their health. In addition, lack of PA has been linked to high levels of stress in older adults, whereas those older adults who are physically active report less stress.[38] Similarly, PA has been identified as a cost-effective, efficient and non-invasive way to

reduce stress in older adults and to enhance their quality of life.[39]

Evidence also supports a link between PA and reduced levels of depression in older adults. For instance, in their study including 1123 Brazilian older adults (aged ≥60 years),Lage et al [40] identified that approximately 83.8% of participants demonstrated a decline in PA level, and that there is an inverse relationship between PA and depression. Participants who were engaged in daily moderate PA (β = −0.174; 95% CI = −0.026 o −0.012) and moderate-to-vigorous PA (β = −0.183; 95% CI = −0.023 to −0.011)[40] reported lower levels of depression than those who were not physically active. These findings corroborate a substantial body of prior evidence, including older adult participants and longitudinal explorations.[41–43] Further Rethorst et al also observed in women aged 18–74 years that substituting SB with vigorous PA was related to a reduction in depressive symptoms, but this was not the case for light or moderate intensity PA.[44]

A significant association has also been uncovered between higher levels of anxiety and lower levels of self-reported PA in older adults.[42] Reviews have indicated that moderate planned PA is strongly related to a decreased anxiety level.[45] A nested case–control study with older community participants identified that those completing a high level of PA were less likely to suffer from cognitive impairment (OR 0.58, 95% CI 0.41 to 0.73).[46] Moreover, the combined evidence from different reviews signifies that PA might improve cognitive process as well as prevent cognitive impairment among older adults thus improving older people's quality of life through a range of mechanisms.[47 48] Therefore, encouraging older adults to be active and independent is a public health priority.

The UK guidelines on PA for adults suggest performing at least 150 min per week of moderate-intensity PA or 75 min per week of high-intensity PA.[49] However, it has been reported that only 30% of older adults aged ≥75 and 55% of older individuals aged 55–74 meet the PA standards, as compared with 83% of adults in the age range of 18–24 years.[50] This decline in PA among older adults has been attributed to numerous factors that include limited knowledge regarding PA benefits, lack of access to PA facilities, and lack of resources, time and support.[51–53] Literature highlights that housing is a major contributing factor to health, and better housing condition has been linked to improved general and psychological health.[54 55] In contrast, some advancements in the home setting, like the increased number of electronic media and devices, have reduced PA levels and enhanced sedentary behaviour, leading to detrimental consequences for public health.[56] There has been strong evidence that the physical environment within the home space has a substantial impact on sedentary behaviour and PA in children.[57] However, there is a lack of research that has investigated the physical setting within the home environment in the context of older adults' PA level and sedentary behaviour. Given that with increasing age, older adults spend large proportions of their time in their houses,[58]

it is important to optimise older adults' home settings to support healthy ageing, including engagement in PA and reductions in SB.

Moreover, in the current COVID-19 pandemic, there is lack of access to community services and activities that has significantly affected older adults' life, with many finding it difficult to manage the repercussions of the pandemic, such as increased fear of being outside, and return to life as lived prior to the pandemic.[59] Being home bound has led many older adults to spend their time watching television. A telephonic poll has showed that 73% of the older adults claimed that during pandemic they watched more television as compared with before the COVID-19 pandemic. Some have reported feeling guilt due to spending more time watching television or playing videos and many older adults faced difficulties in maintaining their well-being, health and independence.[59] The pandemic has therefore had a detrimental impact on their life in diverse ways by limiting their participation in different activities that are beneficial for their well-being.[60] Older adults have experienced negative effects of staying at home, being isolated, feeling a lack of confidence and being scared about returning back to normal life.[61] This has resulted in severe consequences for their physical and mental health and impaired their ability to independently perform their daily living activities. A survey showed that 42% of the older adults (>60 years) in the UK find it harder to walk up and down stairs and to walk short distances. Thirty-six per cent find it extremely challenging to take a shower or have a bath or wash whereas 43% of them were unable to prepare and cook food.[60] Therefore, there is a need to enhance older adults' PA level within their home. This will encourage a healthy lifestyle among older people which in turn will increase their quality of life and foster their active participation in communities, reducing the personal and societal burden of physical inactivity and associated ill health.[62] Moreover, little is known about older adults' perceptions and use of their home environment for reducing sedentary behaviour and increasing PA. Therefore, the current study aims to explore older adults' perceptions around optimising their physical environment within their home space to support PA.

## METHODS AND ANALYSIS
### Study design
An exploratory qualitative research design using semi-structured interviews and a purposive sampling approach will be employed in this study from 1 August 2022 to 31 January 2023. In-depth interviews (IDIs) with older adults will be used as a data collection method for this qualitative research. The IDIs aim to explore perceptions of older adults towards optimising their physical environment within their home space to support PA. This study will be reported according to the guidelines for qualitative research provided in the consolidated checklist (online supplemental file 1).

## Study setting and study participants

The study will be conducted with older adults of different genders who will be selected based on the eligibility criteria (see Eligibility criteria). In the first stage of participant recruitment different community organisations in Swansea, Bridgend and Neath Port Talbot will be approached to formally request approval to recruit via their contacts. Regional coordinators and subcoordinators will act as gatekeepers as they will be approached for recruitment, to best ensure participating older adults reflect the typical socio-demographics of South Wales. Further, advertisement across the University using the intranet and work alongside the city and county of Swansea, Neath Port Talbot and Bridgend Borough Council will be done to recruit using their sport, play and community networks. Recruitment letters, posters/leaflets will be used as an enrolment technique.

The lead researcher will request approval from the project leads to make a visit to pitch for participants in the study. Interested older adults will be asked to express their interest via their respective area project co-ordinator. The researcher will contact interested older adults via email or phone. Older adults who agree to take part will receive participant information sheets, at which point a time for the researcher to meet them will be arranged that is mutually convenient and where the lead researcher will explain the study process and procedures if necessary. Providing the older adult is happy to proceed, they will be given a consent form to complete to carry out the interview.

## Patient and public involvement

There is no patient or public involvement in setting the research agenda.

## IDIs with older adults

We will conduct IDIs with older adults to explore their perceptions towards optimising their physical environment within their home space to support PA and reduce sedentary behaviour. IDI participants will be identified via the gatekeepers that will be approached by the researcher to seek permission to contact potential older adults via their organisation. All interested participants will be asked to further discuss the study and indicate their interest. The researcher will then organise a time for formal enrolment and induction to the study. All participants will provide written informed consent to participate in the study. The participants will be informed that the interview will be audiorecorded, that written notes might also be taken and they are not required to answer any questions they do not wish to. All IDIs will be carried out in English and are anticipated to last around 30–40 min. Older adults will be informed that their data will remain confidential and there will be no identifying characteristics included on the verbatim-typed transcript. The main themes will consist of a general conversation about older adults' perceptions about optimising their home environment to support PA and to explore the factors within older adults' home spaces that promote PA and well-being and reduce home-based sedentary behaviours. We anticipate that 15–20 interviews will be carried out, however, interviews will be deemed complete based on the saturation of the data and when no new themes emerge from the interviews. Therefore, data collection and analysis will occur simultaneously to determine the point of data saturation.

## Eligibility criteria

The following are the criteria for inclusion and exclusion of study participants:

### Inclusion criteria

► Participants can be male or female living in their home.
► All participants aged ≥65 years old.
► Be able to communicate in English.

### Exclusion criteria

Participants will be excluded from the study if they:
► Are not able to cooperate with the research team for the full duration of the project.
► Experience any physical/psychological issues that may put them at risk.

## Data collection procedure

An IDI guide has been developed for older adults. All older adults will be provided with study information sheets and the study will be clearly explained to them. They will be told that participation is entirely voluntary, and they can withdraw at any point. All consenting participants will be interviewed. Each participant will be given a specific ID number which will be used throughout the study. A background survey will be designed to gather descriptive information about the older adults that includes age, gender, education etc. This will be followed by the semistructured interviews that will allow the researcher to obtain information about the participants' experiences of using and interacting with their home spaces to perform PA, their experiences of spending time in their home and the impact on their PA. With regard to the older adults, we will also be specifically exploring activities they like to do. The interview will explore issues such as: How do they spend time at home? What kind of PA they perform at home? What are the ways of being physically active? They will also be asked about the number of days per week on which they are active for about 150–300 min, and the time they spend watching TV, sitting and using the internet for leisure time on a specific week or weekend day. They will be asked to discuss typical activities they carry out at home. They will be asked about their home environment, including topics such as how they selected their home and how it can be modified to enhance PA and lessen their sedentary behaviour at home. The impact of social isolation and restricted activity will also be explored. All semistructured interviews will be conducted either face to face or online via Zoom or WhatsApp depending on the older adult's preference . Based on the older adult's convenient day and time interviews will be scheduled. Interviews are

anticipated to begin in mid July 2022. The researcher will also keep a reflective diary. This will be used to record informal and incidental comments and observations, reflect on the recruitment and study process (including any feedback received from individuals who decline to participate) and to produce a reflective epilogue.

## Data analysis

First, all collected data will be transcribed verbatim by the interviewer (lead researcher). Reflexive thematic analysis will be done and NVivo V.12 Plus software will be used to import, organise and explore data for analysis. To develop clarification and familiarity with the data two independent investigators will read the transcript several times. We will perform an iterative process that will guide us to categorise data and create new categories to generate emergent themes. The documented text will be separated into short units and labelled as a 'code' that reflects the main themes of interest in the study. Consequently, codes will be analysed and combined into similar categories, and these will be combined into subthemes and final themes based on the socioecological framework. The socioecological approach emphasises that an individual is not separate from the environment in which he or she spends their time, and the physical environment has an influence on the individual's behaviour, particularly PA.[63] [64] Two independent researchers will execute the coding, categories generation and reflexive thematic analysis. To acknowledge researcher bias, disagreements between the two analysts will be settled through consensus sessions, with the second author acting as a critical friend, therefore, challenging each other's interpretations.

## Ethics and dissemination

Prior to participating in the study, older persons will be asked to submit written informed consent. Older adults who are not able to write down their names will be requested to give a thumbprint to signify their willingness to participate. Ethical approval for this study has been obtained from the College of Engineering Research Ethics Committee (NM_31-03-22), Swansea University. The findings of the study will be disseminated to the scientific community in a peer-reviewed journal article and to the study participants via workshops and seminar.

## DISCUSSION

The physical environment is a modifiable factor, thus, optimising the physical setting for PA is crucial because individual change within the home environment can be an easily accessible way to drive significant changes without a facilitative infrastructure. The physical environment of the household is particularly important to older adults because it often determines opportunities for PA. The socioecological approach emphasises that an individual is not separate from the environment in which she/he spends their time, and the physical environment has an influence on the individual's behaviour,

particularly PA.[64] Moreover, the study findings will help us to understand and improve the geography of older adults' homes to enable them to be physically active and reduce their sedentary behaviour which in turn improves their health. It will help us to create awareness among older adults and widen the scope of the existing research to support the development of proper pathways in optimising the home environment to support PA. Moreover, the results of this study can guide the designing and planning of specialised interventions aimed at encouraging older adults' PA within their home space. Furthermore, the findings of this study must take into account knowledge about what motivates older people to alter their behaviour as well as how to maintain a positive change in behaviour within their home environment to perform PA. Additionally, involving older adults should be a key aspect of any intervention to enhance the physical environment for PA, as older adults could offer invaluable insights into local need and could work together with the practitioners to increase the potential success of interventions. In addition, this research aims to inform social care policy that adopts a preventive healthcare model to maintain older adults' well-being by enabling them to be independent in their home surroundings. This will encourage researchers, designers, developers, healthcare experts and policy makers to work together to identify and provide suitable and supportive housing environments and home spaces for an ageing population.

**Contributors** NAAM designed the protocol under the supervision of JH, GS and JM. JH and GS gave multiple critical feedback on overall draft. JM provided insights from her expertise in catering several research on older adults' population, which added more value in designing and revising protocol. All authors contributed to reviewing and editing the final draft.

**Funding** This is funded by: Economic and Social Research Council (ES/P00069X/1).

**Competing interests** None declared.

**Patient and public involvement** Patients and/or the public were not involved in the design, or conduct, or reporting, or dissemination plans of this research.

**Patient consent for publication** Not applicable.

**Provenance and peer review** Not commissioned; externally peer reviewed.

**ORCID iD**
Naureen Akber Ali Meghani http://orcid.org/0000-0001-5442-5598

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
