## [Reviewer comments · BMJ Open]

ARTICLE DETAILS

TITLE (PROVISIONAL)	Optimizing older adults home spaces to enhance their physical activity level: An exploratory qualitative study protocol
AUTHORS	Meghani, Naureen; Hudson, Joanne; Stratton, Gareth; Mullins, Jane

VERSION 1 – REVIEW

REVIEWER	Ghram, Amine University of Tehran Faculty of Physical Education and Sport Science
REVIEW RETURNED	24-Sep-2022

GENERAL COMMENTS	I have no specific concerns and no additional comments and I congratulate the authors for their study protocol. All abbreviations should be explained the first time they are used - unless it is a standard unit of measurement - and thereafter the use of abbreviations should be consistent throughout the paper
---

REVIEWER	Sommantico, Massimiliano University of Naples Federico II, Department of Humanities
REVIEW RETURNED	26-Oct-2022

GENERAL COMMENTS	Thank you for the opportunity to review this manuscript (bmjopen-2022-066940) examining the relationship between older adults' home spaces and their physical activity level. A particular strength of the manuscript is its potential to address an important gap in the literature by examining an understudied population. As such, the findings have the potential to make a contribution to the literature and inform public health practice. The background and justification for the study is thoroughly discussed and engrained within the academic literature. My principal concerns regard the cited literature, especially in different cultural contexts, and some minor revisions. 1. I suggest improving the introduction and rationale of the study not immediately focusing on physical activity, but first introducing the question of meaning in life among the elderly, especially related to home space meaning, also in different cultural contexts. Here is some suggested references: Chang, S. O., & Patricia, M. B. (2000). Meaning in life among the elderly. Journal of Korean Academy of Nursing, 30(2), 259-271; Hofer, J., Busch, H., Au, A., Poláčeková Šolcová, I., Tavel, P., & Tsien Wong, T. (2014). For the benefit of others: generativity and meaning in life in the elderly in four cultures. Psychology and Aging, 29(4), 764-775; Lacatena, M., Sommantico, M. (2022). Old age, time, and the space of the home: A qualitative research study during the COVID-19 pandemic. Mediterranean Journal of Clinical Psychology, 10(2), 1-
--

	22; Takkinen, S., & Ruoppila, I. (2001). Meaning in life in three samples of elderly persons with high cognitive functioning. The International Journal of Aging and Human Development, 53(1), 51-73; 2. I suggest moving the Strengths and limitations section before the discussion; 3. I suggest explicit which theory guided the analyses of the interviews (Grounded Theory?); 4. I suggest deepening the Discussion section.
--	---

VERSION 1 – AUTHOR RESPONSE

S. No.	Reviewer 1 Comments	Point by point response
	Dr. Amine Ghram, University of Tehran Faculty of Physical Education and Sport Science, Tehran University of Medical Sciences Comments to the Author: I have no specific concerns and no additional comments and I congratulate the authors for their study protocol. All abbreviations should be explained the first time they are used - unless it is a standard unit of measurement - and thereafter the use of abbreviations should be consistent throughout the paper	Thank you for your positive feedback

S. No.	Reviewer 2 Comments	Point by point response
	Dr. Massimiliano Sommantico, University of Naples Federico II Comments to the Author: Thank you for the opportunity to review this manuscript (bmjopen-2022-066940) examining the relationship between older adults' home spaces and their physical activity level. A particular strength of the manuscript is its potential to address an important gap in the literature by examining an understudied population. As such, the findings have the potential to make a contribution to the literature and inform public health practice. The background and justification for the study is thoroughly discussed and engrained within the academic literature.	Thank you for your positive feedback.

1	My principal concerns regard the cited literature, especially in different cultural contexts, and some minor revisions I suggest improving the introduction and rationale of the study not immediately focusing on physical activity, but first introducing the question of meaning in life among the elderly, especially related to home space meaning, also in different cultural contexts. Here is some suggested references: Chang, S. O., & Patricia, M. B. (2000). Meaning in life among the elderly. Journal of Korean Academy of Nursing, 30(2), 259-271; Hofer, J., Busch, H., Au, A., Poláčková Šolcová, I., Tavel, P., & Tsien Wong, T. (2014). For the benefit of others: generativity and meaning in life in the elderly in four cultures. Psychology and Aging, 29(4), 764-775; Lacatena, M., Sommantico, M. (2022). Old age, time, and the space of the home: A qualitative research study during the COVID-19 pandemic. Mediterranean Journal of Clinical Psychology, 10(2), 1-22; Takkinen, S., & Ruoppila, I. (2001). Meaning in life in three samples of elderly persons with high cognitive functioning. The International Journal of Aging and Human Development, 53(1), 51-73;	Thank you for your suggestions. Based on the reviewer feedback, suggested revisions have been made in introduction section on page no. 4, line no. 2 -25 and page no. 5 line no.1-19
2.	I suggest moving the Strengths and limitations section before the discussion;	Thank you for your feedback. The editor has suggested to follow this format.
3.	I suggest explicit which theory guided the analyses of the interviews (Grounded Theory?);	Thank you for your comments. We have added suggested part in the analysis section. Changes are done on page no. 14 line no. 12-17
4.	I suggest deepening the Discussion section.	Thank you for your valuable feedback. We have strengthened the discussion part. Changes are done on page no. 15 and line no. 5-11 & 16-24 and page no. 16 line no. 1-3